

# Development and implementation of a new biomass burning emissions injection height scheme for the GEOS-Chem model

Liye Zhu[1], Maria Val Martin[2,3], Arsineh Hecobian[1], Luciana V. Gatti[4], Ralph Kahn[5], and Emily V. Fischer[1]

[1] Department of Atmospheric Science, Colorado State University, Fort Collins, CO, USA.
[2] Chemical and Biological Engineering Department, The University of Sheffield, Sheffield, UK.
[3] Now at Leverhulme Center for Climate Change Mitigation, Animal Plant Sciences Department, The University of Sheffield, Sheffield, UK.
[4] Instituto de Pesquisas Energéticas e Nucleares (IPEN)–Comissao Nacional de Energia Nuclear (CNEN)–Atmospheric Chemistry Laboratory, Cidade Universitaria, Sao Paulo CEP, Brazil.
[5] Climate and Radiation Laboratory, NASA Goddard Space Flight Center, Greenbelt, Maryland, USA.

Correspondence to: Emily V. Fischer (evf@atmos.colostate.edu)

**Abstract.** Biomass burning is a significant source of trace gases and aerosols to the atmosphere, and the evolution of these species depends acutely on where they are injected into the atmosphere. GEOS-Chem is a chemical transport model driven by assimilated meteorological data that is used to probe a variety of scientific questions related to atmospheric composition, including the role of biomass burning. This paper presents the development and implementation of a new global biomass burning emissions injection scheme in the GEOS-Chem model. The new injection scheme is based on monthly gridded Multi-Angle Imaging Spectro Radiometer (MISR) global plume-height stereoscopic observations in 2008. To provide specific examples of the impact of the model updates, we compare the output from simulations with and without the new MISR-based injection height scheme to several sets of observations from regions with active fires. Our comparisons with ARCTAS aircraft observations show that the updated injection height scheme improves the ability of the model to simulate the vertical distribution of peroxyacetyl nitrate (PAN) and carbon monoxide (CO) over North American boreal regions in summer. We also compare a simulation for October 2010 and 2011 to vertical profiles of CO over the Amazon Basin. When coupled with larger emission factors for CO, a simulation that includes the new injection scheme also better



matches selected observations in this region. Finally the improved injection height also improves the simulation of monthly mean surface CO over California during July 2008, a period with large fires.

**1 Introduction**

Properly describing the injection altitude of smoke in the atmosphere is an essential step in predicting the impact of emissions from landscape fires on atmospheric composition (Paugam et al., 2016). Injecting smoke higher in the atmosphere in chemical transport models can extend or reduce lifetime of trace species, and it can alter the spatial

extent of smoke-influence in the atmosphere (Freitas et al., 2006). The impact of injection height on smoke dispersion is three-fold: 1) winds in the free troposphere are generally stronger than in the boundary layer, thus when smoke is emitted aloft, defined plumes can be detected thousands of kilometers downwind (e.g., Colarco et al. (2004); Damoah et al. (2004); Forster et al. (2001); Val Martín et al. (2006)). 2) Removal processes tend to be

more efficient in the boundary layer (e.g. Boy et al. (2008)). 3) Chemical evolution within the plume can be sensitive to injection height since altitude impacts plume temperature, ambient relative humidity, smoke-cloud interactions, and photolysis rates (e.g. Freitas et al., 2006). Given the importance for atmospheric composition and air quality predictions (e.g. Stein et al., 2009), substantial efforts have been made to better

understand how injection height varies by ecosystem type and season (e.g., Val Martin et al. (2010); Tosca et al. (2011); Mims et al. (2010)), which environmental drivers of injection height are most important (e.g., Kahn et al. (2007); Val Martin et al. (2012)), and how best to estimate smoke injection height in models (e.g.,  Paugam et al. (2016) and references therein) to produce improvements in model simulations of trace

constituents (e.g., Gonzi et al. (2015)).

GEOS-Chem is a global chemical transport model (CTM) (www.geos-chem.org) (Bey et al., 2001) that is routinely used to simulate the impacts of biomass burning on atmospheric composition (e.g., Lewis et al. (2013) and Leung et al. (2007)). GEOS-Chem is driven by GEOS assimilated meteorological data from the NASA Global Modeling and

Assimilation Office (GMAO), and it includes a state-of-the-science description of tropospheric oxidant chemistry, necessary for understanding the chemical and dynamical processes controlling the evolution of biomass burning emissions. The public release



version of GEOS-Chem emits all biomass burning emissions into the atmospheric
boundary layer. This may be appropriate for some fire types, but is likely a source of
error for many regions with active biomass burning (e.g. Leung et al., 2007). The main
objective of the current paper is to introduce a new global biomass burning injection
height scheme for GEOS-Chem based on Multi-Angle Imaging Spectro Radiometer
(MISR) plume injection height observations from 2008. Smoke aerosol injection height is
derived from source plumes with discernable features in the MISR multi-angle views
(Kahn et al., 2008).

        Though biomass burning impacts atmospheric composition across a suite of
temporal and geographic scales, this paper presents model-observation comparisons for
specific biomass burning plumes having well-sampled vertical structure. The data
available to make such important comparisons is limited. However, this is an important
step toward using the model to address broader aspects of atmospheric composition. To
the best of our knowledge, this paper represents the first effort at using measured global
smoke plume injection heights from MISR as constraints on a CTM. There have been
efforts to do this on a regional scale for specific fire seasons (e.g., Chen et al., 2009; Jian
and Fu, 2014), but we are unaware of similar global implementations. Val Martin et al.
(2012) studied the performance of one of the most advanced physically based plume-rise
models. They concluded that given the uncertainties and performance of that approach,
empirically derived plume injection heights, such as those we use here, provide better
constraints on smoke transport.

        Much of the model development presented here was motivated by the persistent
challenge CTMs appear to face at accurately simulating peroxyactyl nitrate (PAN) in the
atmosphere (e.g. Emmons et al. (2015)). This compound plays a central role in oxidant
chemistry, particularly in remote regions (Moxim et al., 1996). However, it has a
temperature dependent lifetime (Singh and Hanst, 1981), and thus its evolution in the
atmosphere is particularly sensitive to plume injection height. As a first step toward
validating the revised model, we compare the output from a simulation with improved
injection heights to multiple sets of observations from regions with active fires, providing
examples of cases where injecting a substantial percent of biomass burning emissions in



the free troposphere is important for properly simulating this trace species as well as carbon monoxide (CO).

## 2 Methods

### 2.1 Overview of Model Development

Figure 1 illustrates the process of implementing an observationally based injection scheme into GEOS-Chem. This section describes the details associated with each step in the process. The new injection scheme is based on MISR plume injection height observations from 2008 (Section 2.2). We then map the native MISR injection altitude (0-8 km) to emitted percentages of total biomass burning emissions to GMAO 47-layer reduced vertical grid and $2^o \times 2.5^o$ horizontal grid (Section 2.3). The model configuration is described in Section 2.4.

### 2.2 Analysis of MISR Plume Height Observations

The new injection scheme is developed based on the MISR plume-height stereoscopic observations in 2008 (Val Martin and Kahn, in prep). The MISR data we used is part of the MISR Plume Height Project2, which was derived for the AeroCom multi-model biomass burning experiment. The dataset is publicity available from https://misr.jpl.nasa.gov/getData/accessData/MisrMinxPlumes2/. We developed a plume-height parameterization from statistical summaries of worldwide, region-specific MISR plume height retrievals. The parameterization consists of fire emission percentages based on land cover units, stratified by altitude, region and season. For the land cover classes, we used the land cover map from the annual MODIS MCD12C1 land cover type product (Friedl et al., 2010). The vertical resolution of the injection height is 250 m, extending from 0 to 8 km. An overview of the MISR instrument and standard products is given by Diner et al. (1998), and more details about the MISR plume digitizing tool and the MISR plume database can be found in Nelson et al. (2013) and Val Martin and Kahn (in preparation), respectively.

There are several subtleties to the MISR-based plume-height climatology that are worth specifically noting here. MISR equator-crossing time during the day is about 10:30 AM, so the diurnal distribution of emissions is not sampled, and in particular, the mid-late afternoon, when wildfires tend to be most intense. Also, for several reasons, the MISR-based plume-height climatology does not include plumes smaller than a certain



size, and this size varies with observing conditions. Several factors contribute to this

limitation. MODIS thermal anomalies are used to identify fire locations, some fires are
smaller than MODIS pixels, others can be obscured by the tree canopy or overlying
smoke, and fires for which the emissivity at 4 microns is low (e.g., smoldering fires), are
sometimes missed (Kahn et al., 2008). These issues also affect satellite-based smoke
emissions inventories such the one used here (see Section 2.4). To acknowledge this

issue, we include an adjustment to the smoke injection height scheme to account for
small fires. Specifically, we use GFED4s (Randerson et al., 2012) to estimate the fraction
of small fires in each region and biomes in 2008. As nearly all small fires inject smoke
only within the boundary layer, we apply a small-fire correction to the lowest model
atmospheric layer as described in Val Martin and Kahn (in preparation). Note that aside

from the small-fire information in GFED4s, derived separately from the standard satellite
retrieval approach of the GFED products, we use GFED version 3 for this study.  The
emission factors for several species, such as CO for Temperate Forests, are lower in
GFEDv4 compared to GFEDv3, which exacerbates known problems of low CO with
GFED-initialized models (Akagi et al. 2011; van der Werf et al. 2017).  As discussed in

Section 2.4 below, we increased and tested the emission factors in GFEDv3 based on the
findings in Petrenko et al. (2017).

**2.3 GEOS-Chem Implementation**

We map the native MISR injection altitude (0-8 km) to emitted percentages of
total biomass burning emissions to the GMAO 47-layer reduced vertical grid and a $2^o \times$

$2.5^o$ horizontal grid. The injection percentages of total column biomass burning emissions
for each month for each grid cell are saved in a binary file. Code modifications to read in
the data of percentages and distribute the biomass burning emissions to every grid cell are
contained within the setemis.F FORTRAN module in GEOS-Chem version 9.01.01. The
binary file could be updated based on different analyses (i.e. a more recent year, or a

different analysis approach), and little effort would be required to update this within the
code.

In Figure 2a, we show an example of vertical profile of injection percentages at
$56^oN$, $105^oW$ from the standard model and the new injection scheme. In contrast to a
blanket approach of emitting all biomass burning emissions within the boundary layer,



the new injection scheme emits a large percent of these emissions above the boundary

layer at this location. The global map in Figure 2b shows the injection percentages at 850

hPa in July 2008 for the globe based on the MISR stereo-height data. The amount of the

total biomass burning emissions at any given location that are injected into the layer

encompassing 850 hPa varies substantially. Figure 2a cannot be interpreted as the total

amount of smoke emitted in this layer of the atmosphere; this is a plot of the percent of

the total column that the model will emit at that level. There are regions during the month

of July 2008 with high percentages of emissions injected at a given level, but very small

total column emissions overall. As discussed in the preceding and following sections, our

parameterization is likely weakest for locations where emissions are dominated by small

fires.

**2.4 GEOS-Chem Configuration**

We use the Goddard Earth Observing System-Chemistry (GEOS-Chem) global 3-

D chemical transport model including detailed ozone-$NO_x$-VOC-aerosol chemistry

(version 9.01.01, www.geos-chem.org) with modifications to emitted species and the

chemical mechanism specifically for PAN as described in Fischer et al. (2014). Most

relevant to this work, we use Global Fire Emissions Database (GFEDv3) monthly

biomass burning emissions (van der Werf et al., 2010), with updated emission factors for

non-methane volatile organic compounds (NMVOCs) and nitrogen oxides ($NO_x$) from

Akagi et al. (2011).  The current work aims at addressing specifically the issue of

injection height. Our injection height parameterization could be used with any emission

inventory. The version of GEOS-Chem that we chose for implementing the improved

injection height scheme is probably the best available in terms of PAN chemistry.

Choosing a monthly-averaged emission dataset can create biases for specific case studies

of biomass burning. However, we have used the standard input file setting used in GEOS-

Chem; this setting is used in benchmarks.

Given the combined limitations in the MISR analysis and the GFED emissions

database at representing small fires, our scheme is unlikely to correctly represent the

fraction of total smoke that is emitted above the boundary layer in places where small

fires make a significant smoke contribution. Randerson et al. (2012) updated the GFEDv4

inventory to include an estimate of the emissions from fires below the detection limit of



the satellite observations used to construct the standard GFED database, and most other satellite-based emission inventories. These small fires tend to include agricultural and shrubland fires as well as some grassland fires, peat fires, and ground fires where the overlying tree canopy is dense. The number of small fires is large in some places, their

overall contribution to total emissions can be large, and they often produce diffuse, smoky haze rather than discrete plumes. They also tend to inject smoke into the planetary boundary layer rather than above it. These fires are not the focus of the MISR injection height analysis or the MODIS FRP analysis, and although we have attempted to account for this (Val Martin and Kahn, in preparation), this is a limitation of our overall approach.

PAN in biomass burning plumes is particularly sensitive to injection altitude because the lifetime of PAN is highly temperature dependent. Thus we focus a substantial portion of our model-measurement comparison on this species. For the comparisons we present here, we remove the increased biomass burning emissions for northern Asia, originally applied for 2008 in Fischer et al. (2014). These were applied in Fischer et al.

(2014) because Kaiser et al. (2012) and Yu et al. (personal communication) found that GFEDv3 underestimates fire emissions at boreal latitudes. Given the model experiments we applied following Petrenko et al. (2017) (see below), we removed this additional increase from the Fischer et al. (2014) configuration. We also remove the injection partitioning assumption in Fischer et al. (2014), which emitted 35% of total biomass

burning emissions above the boundary layer to test the sensitivity of PAN to this choice. Fischer et al. (2014) found this to improve the PAN simulation, but this is a much coarser approach than what has been done here. The model experiments in Fischer et al. (2014) were among main motivations for the current paper. In the following text and figures, we refer to this version of model as the "standard model" because the injection of biomass

burning is treated as in the public release benchmarked version of GEOS-Chem. We refer to the observationally based injection scheme as the "new injection scheme." As a final model experiment, we increase the CO emissions by a factor of 1.5 for burning over savannas, a factor of 1.5 for burning associated with deforestation, and a factor of 2 for extra-tropical forests following Petrenko et al. (2017). We use $2° \times 2.5°$ horizontal

resolution for our global simulations.

**2.5 Observational Datasets**



As a demonstration of the potential impact of the model development and its
relevance to a few example regions, we compare GEOS-Chem output with improved
injection heights to smoke-impacted trace gas observations from aircraft over boreal

North America (July 2008) and from aircraft sites in the Amazon basin (2010 – 2011).
We also compare the model output to monthly mean surface CO observations in regions
impacted by major fires.

### 2.5.1 North America

Boreal North America is an interesting focal region because emissions from

biomass burning lead to enhancements in high latitude tropospheric ozone during
summer (Arnold et al., 2015). The representation of injection height has implications for
inverse studies of emissions from fires in this region and the magnitude of the ozone
enhancement that results from these emissions (Leung et al., 2007). The second portion
of the NASA Arctic Research of the Composition of the Troposphere from Aircraft and

Satellites (ARCTAS) mission was conducted over western Canada during June and July
2008. A complete list of species observed by the NASA DC-8 aircraft during ARCTAS
can be found in Jacob et al. (2010). In the present study, we use ARCTAS observations
of carbon monoxide (CO) and PAN from July 2008 to illustrate the updated performance
of the model with the new injection scheme over western North America. To synchronize

data used in the model, the timing of all data was adjusted to UTC time following
Hecobian et al. (2011).

### 2.5.2 Amazon

We highlight the Amazon basin as another interesting region as emissions from
deforestation fires over Amazonian forests represent a large percent of global emissions

from deforestation (van der Werf et al., 2010). Year-to-year variability in this region has
been associated with climate extremes (Chen et al., 2013). Thus, it is important to
understand the fire injection height over this region in order to fully quantify the impact
of these fires on atmospheric composition, and to better predict how this impact could
evolve in the future. We use the CO observations from four sites across the Amazon

basin in 2010 and 2011: Alta Floresta (ALF; 8.80ºS, 56.75ºW), Rio Branco (RBA;
9.38ºS, 67.62ºW), Santarém (SAN; 2.86ºS; 54.95ºW) and Tabatinga (TAB; 5.96ºS,
70.06ºW). Bi-weekly vertical profiles of CO were measured from just above the forest





canopy to 4.4 km above sea level (Gatti et al., 2014). Thus, up to six or eight observations were available at one altitude level in one month (4 sites with 2 vertical profiles). The

measurements were taken at specific altitude levels for each measurement day to get profiles of CO mixing ratios. We also compared model output to aircraft observations from the Balanço Atmosférico Regional de Carbono na Amazônia (BARCA) program, which deployed in 2008 (Andreae et al., 2012). This dataset contains a strong influence of biomass burning emissions. However, when we sampled the model at the locations of

the observations, there were no differences in the simulated CO profiles in the two sets of simulations with the different injection schemes. The CO mixing ratios for the regions were biased low, i.e. model mixing ratios were between 80 and 125 ppb, indicating no smoke influence, whereas the corresponding observations were largely > 150 ppb. Andreae et al. (2012) discuss problems with GFEDv3 CO emissions for this region;

specifically noting that the emissions in this database could be up to a factor of seven too low for the BARCA period.

## 3 Results

### 3.1 North American Boreal Fires

Figure 3a shows the total percent of biomass burning emissions for each model

column emitted above 700 hPa during July 2008 based on MISR observations. We use the 700 hPa level to signify an approximate mid-day boundary layer top pressure over North America. The percent of emissions injected above 700 hPa in the updated version of the model is quite large over boreal regions, exceeding 60% for some locations such as the one shown in Figure 3b. In boreal regions, the majority of biomass burning emissions

are produced by a relatively small number of large fires that last days to weeks (Stocks et al., 2002; Brey et al., 2018). Figure 3a shows the strong north-south gradient in the percent of emissions injected above this atmospheric level. In contrast to boreal regions, the new scheme continues to inject nearly all the fire emissions into the boundary layer over the central U.S. during this month. An example profile of the emitted percent by

model layer is shown in Figure 3c. There are typically very few fires during July in this region; those that do occur are typically short-lived and often involve cropland (Brey et al., 2018).

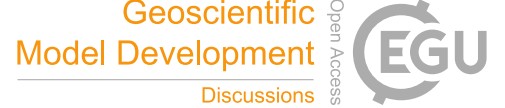



The impact of the new injection scheme on simulated PAN has significant spatial variability over North America during July 2008, and this is driven by the large spatial variability in the fires and the smoke injection level. Figures 4a - 4c present the differences in simulated PAN mixing ratios between the updated and the standard model at the surface, 850 hPa and 510 hPa on 1 July 2008, respectively. As expected, the new injection scheme decreases simulated PAN mixing ratios at the surface and within the boundary layer over boreal regions. Simulated PAN mixing ratios increase in the mid and upper-troposphere. Figure 5 presents a similar example for 4 July 2008.

Figures 4d and 5d (black lines and open circles) show average vertical profiles of PAN intercepted by the DC8 during the ARCTAS flights on these particular days. The NASA DC8 sampled fresh smoke from the Lake McKay fire (56.5°N, 106.8°W) on 1 July 2008 at several distances downwind (see Alvarado et al. (2010)). We sampled both versions of the model along the aircraft pathway at the corresponding observation time, and these average profiles are also plotted in Figures 4d. In the lower troposphere, the standard model largely overestimates PAN on 1 July 2008 (Figure 4d). The new injection scheme decreases the simulated PAN in the boundary layer significantly (~ 300 pptv) and matches better to the ARCTAS observations. The DC8 sampled several plumes above 3 km on 4 July. As described in Alvarado et al. (2010), this was a period with strong updrafts, which led to lofting of biomass burning emissions (Fuelberg et al., 2010). We note that time of day could be very important for these comparisons. The aircraft sampled these plumes in the mid-to-late afternoon, so MISR heights are likely underestimates of the actual injection altitudes for the cases shown in Figures 4, 5, and 6.

On 4 July 2008 (Figure 5d), the new injection scheme does not change simulated PAN meaningfully near the surface where the aircraft was located. Figures 5e and 5c show that the aircraft did not fly through the low altitude regions of the model, which showed important changes from the injection scheme. The near surface PAN mixing ratios were not impacted south of the Hudson Bay. However, the new injection scheme does increase PAN by ~130 pptv in the lower to mid free troposphere. This improves the model-measurement comparison substantially between 800 and 500 hPa. The GEOS-Chem simulations presented in Alvarado et al. (2010) substantially underestimated PAN relative to the ARCTAS observations. Here we have included the partitioning of $NO_x$





immediately to PAN and HNO$_3$ as originally suggested by Alvarado et al. (2010) and we

have updated the injection height. Along with the other updates in Fischer et al. (2014), this appears to greatly improve the ability of the model to simulate the appropriate magnitude of PAN for the cases shown.

Figure 6 presents simulated and observed CO from the 4 July ARCTAS flight. Similar to PAN, for this particular profile the new injection scheme decreases CO in

lower troposphere and increases it in middle troposphere (Figure 6d). However, both the standard model and the new injection scheme underestimate CO significantly compared to ARCTAS observations. The mean CO underestimate shown in Figure 5d is 15%-56%. Alvarado et al. (2010) and Fisher et al. (2010) previously compared a GEOS-Chem simulation to ARCTAS observations. The simulation used in those prior studies was

based on daily emissions from the Fire Locating and Monitoring of Burning Emissions (FLAMBE) emission inventory (Reid et al., 2009). Monthly mean GFEDv2 emissions were used for the model spin up. The FLAMBE inventory overestimated CO emissions from fires in this region (Alvarado et al., 2010). In contrast, we find that CO is under-predicted using GFEDv3 monthly average emissions.

We do not aim to optimize the ability of the model to simulate these specific, plumes, but rather to show the magnitude of the changes with respect to this well-studied set of plumes. Chen et al. (2009) found that switching from monthly to 8-day time intervals for GFEDv2 in GEOS-Chem had the largest effect on simulating measured day-to-day variability in CO for boreal fires during the 2004 fire season. So it is possible that

this approach (or alternatively using daily or 3-hourly fire fractions) would also improve the ability of the model to capture these specific plumes. We did not pursue these options. However, to simply show the impact of changing the emission factors, we include an additional simulation with both the updated injection scheme and increased emissions of CO (factor of 2 for extra-tropical fires and 1.5 for savannahs) following Petrenko et al.

(2017). This final simulation (pink line in Figure 6e) increases simulated CO at all atmospheric levels. The model most closely matches observations between 900 and 800 hPa, but continues to under-predict CO higher up in the atmosphere.

It is important to note that the MISR plume heights that form the basis for our injection scheme are only snapshots. MISR is a sun-synchronous orbit, and it crosses the



equator at 10:30 local time. Actual injection heights vary diurnally and less predictably

hour-to-hour or day-to-day as burning progresses.  Our scheme provides one consistent

injection height for each month; however, the ARCTAS aircraft observations also

represent daytime measurements. A future development may be to attempt to anchor the

model plume height at the MISR overpass time, rather than assuming a constant plume

height. A better comparison would include plumes observed throughout the diurnal cycle.

**3.2 Amazon Basin Comparison**

As discussed above in Section 2.5.2, we compare the simulated CO profiles to

observed CO profiles at four Amazon basin sites in each month during 2010 and 2011.

We note that we evaluated GEOS-Chem over the Amazon with observations collected in

different years than the MISR plume height data used to develop the parameterization.

We made this choice because 2010-2011 CO profiles are available for use in the model-

measurement comparison, and the MISR smoke-plume-height climatology from 2005-

2012 shows little inter-annual variability over this region (Gonzalez-Alonso et al., in

preparation). Where the data and the model are clearly smoke-impacted, a simulation that

includes both the new injection scheme and increased CO emissions does improve the

simulated CO profiles over this region. Figure 7 shows the total emitted percent of

biomass burning emission injected above 700 hPa over the Amazon from March to

November based on the MISR data. From March to August, the total emitted percent in

each grid cell above 700 hPa over the Amazon area is generally <15%. Thus, the new

injection scheme does not produce a large difference in simulated CO profiles compared

to the baseline simulation. However, the total emitted percent above 700 hPa in each grid

cell is generally >25% from September to November. Figure 8b shows the vertical

profiles of the emitted percent of biomass burning smoke from the standard model and

the new injection scheme at the RBA site, for one case in October. Although peak

emitted percentages in both simulations are near the top of the boundary layer, the new

injection scheme has the emissions pushed sufficiently higher in the atmosphere as to

improve the comparison with existing observations. Figure 8c shows comparison of the

simulations with the corresponding bi-weekly observations at RBA in October of 2010

and 2011. Above the lowermost km, the simulated CO from the standard model generally

under predicts the observed CO mixing ratios. The new injection scheme decreases CO





mixing ratios in the boundary layer (by up to 45 ppb) and increases CO mixing ratios in troposphere (by up to 12 ppb). CO mixing ratios are 20 – 75 ppb lower than the RBA observations. With the increased CO emissions (pink), simulated CO mixing ratios near the surface provide a better match to the observations than the other two model versions.

Away from the surface the simulation including both increased CO and the new injection scheme also performs better that the other two model versions, but the model still under predicts CO mixing ratios in this region of the atmosphere by ~50 ppb.

We also compared the model output to the CO mixing ratio profiles over the other three sites (TAB, ALF and SAN), and the impacts of new injection scheme on CO

mixing ratios at all levels over all three locations is small. Consistent with Andreae et al. (2012), we also found that the simulated CO mixing ratios are generally under-predicted in all months, especially during the biomass burning seasons. For example, the simulated CO mixing ratios are almost three times lower than observations in September at the SAN site. Gatti et al. (2014) found an emission ratio of 72.8 ppb $CO/CO_2$ ppm. For

comparison the emission ratios used in GFEDv3 as implemented in GEOS-Chem are 97.5 and 59.5 ppb $CO/CO_2$ ppm for deforestation and savannas respectively.  It is possible that either the emission factors themselves may be too low in GFEDv3 or there are fires missing from the inventory, so redistributing them in the atmosphere is not sufficient to better understand their impact on atmospheric composition.

**3.3 Averaged Impacts on CO**

As the case studies of individual plumes presented in Sections 3.1 and 3.2 show, injecting the emissions of boreal fires higher in the atmosphere often increases the CO mixing ratio in the mid-troposphere above and directly downwind of the fire. For the example in Figure 6, the new model substantially reduces CO mixing ratios near the

surface and at 850 hPa. There is an increase in CO at 510 hPa directly above and directly downwind of the fire as compared to the standard model (Figure 6a). However, Figure 6a also shows a decrease in CO at 510 hPa over much of the domain. When viewed hemispherically, the net effect of lofting emissions out of the boundary layer is to produce lower average CO mixing ratios in the mid-upper troposphere because the

average lifetime of CO against oxidation by OH is slightly shorter. Annual and globally averaged concentrations of OH increase slightly with altitude from 1000 hPa to 700 hPa



(Spivakovsky et al., 2000). Thus when a fraction of the CO emissions are immediately moved out of the boundary layer, this fraction reacts more quickly with OH than in the standard simulation. In our model, the resulting changes to monthly mean CO are not

large, but there is quite a bit of variability by season and location. Typical monthly mean decreases in CO mixing ratios away from freshly injected biomass burning plumes are typically <5% in the mid to upper troposphere. The changes in CO between model versions reflects changed injection heights throughout the Northern Hemisphere, not just the fire producing the particular smoke sampled by the aircraft this day. The response of

CO is very different than that of PAN. The main loss of PAN is via thermal decomposition, so injecting PAN (or its precursors) higher in the atmosphere will increase PAN in the mid-to-upper troposphere. Though we highlight CO and PAN here as examples, injection height will impact the chemical evolution of nearly all species emitted from fires.

Leung et al. (2007) showed that the choice of injection height for boreal fire emissions impacts the simulation of surface CO mixing ratios in the Northern Hemisphere. They compared GEOS-Chem simulated anomalies in CO mixing ratios with surface measurements from the NOAA Earth System Research Laboratory (ESRL) Carbon Cycle Cooperative Air Sampling Network . Therefore, we also performed a

comparison with monthly mean observations for 2008. In most locations where we conducted comparisons, the model with the MISR-based injection height did not produce notably different surface monthly mean CO mixing ratios (not shown). However, there are several stations where the updated model produces substantially lower monthly mean surface CO mixing ratios than the standard model, and this change produces a better

simulation of CO at these locations. Figure 9 shows a comparison of our different model versions to monthly mean surface CO mixing ratios from four sites where there are substantial changes in 2008 monthly mean simulated CO with the new injection scheme. The decreases in simulated surface CO can be substantial when the emissions are moved up higher in the atmosphere based on the MISR analysis. Figure 9a and 9b indicate that

the standard model over predicts July 2008 surface CO mixing ratios at two California monitoring sites, Trinidad Head and Point Arena. There were hundreds of wildfires in Northern California in June and July 2008 (Gyawali et al., 2009; Brey et al., 2018). The



model with the improved injection height parameterization removes a large CO peak in
July that is clearly not present in the surface observations. The lower panels of Figure 9
indicate that the model over predicts surface CO abundances during much of the year at
these two sites in the Southern Hemisphere, Bukit Kototabang (BKT), Indonesia and
Cape Grim (CGO), Tasmania. However, the updated version of the model does reduce
the model-measurement discrepancy at BKT between March and September 2008 by
~50%.

**4 Summary**

This paper introduces the development and implementation of a new global
biomass burning emissions injection scheme in the GEOS-Chem model. The injection
scheme is based on a MISR climatology for 2008. Additional (i.e. based on other
datasets) or updated (i.e. other years) gridded climatologies of injection height could be
implemented with relatively little effort given the code infrastructure that is now in place.
We have completed multi-year simulations with the new injection scheme and compared
the model output to three smoke-impacted observational datasets.

Based on MISR snapshots, the percentage of total column biomass burning
emissions that are typically injected above the boundary layer is relatively high for North
American boreal regions. We find that the updated model is better able to simulate
daytime observed vertical profiles of PAN and CO over boreal regions during the 2008
summer fire season, and including a better representation of injection height is likely very
important for predicting the transport and chemical evolution of smoke plumes
originating in this region. However, the version of GEOS-Chem used here has a
persistent low bias in CO throughout the atmospheric column. Though our injection
height climatology is based on observations from 2008, we also used this to simulate
October 2010 and 2011 for the Amazon region. We made this choice because this season
provided access to CO profiles used for model-measurement comparison, and for this
region, smoke injection heights to not appear to vary much interannually.

In testing our model updates, we consistently found that it was important to do
model-observation comparisons on specific biomass burning plumes with well-sampled
vertical structure. When the model is sampled to match observations with less vertical
information (e.g., MOPITT CO or TES PAN retrievals), the differences between the



simulations appeared very small. However, when the model is compared to specific

plumes, an improved injection height does produce notable differences in the simulations
that can have air quality and possibly climate implications (see also Vernon et al. (2018)).
Thus moving forward, we recommend that simulations with improved vertical injection
height schemes for biomass burning plumes be compared to specific plumes, rather than
larger-scale observations.

Though these model developments offer clear improvements under some
situations, limitations in this approach should be noted. Most importantly, the MISR
climatology that underpins this model development is based on snapshots of injection
height Thus it may not apply to all fires in a given location at all times of day.  The MISR
plume height climatology also may not represent the injection height of small fires as

well as it does for larger ones. Thus we expect that this approach will be most appropriate
in regions where the total smoke emissions are dominated by fires large enough to be
observed by the satellite instrument. However, most small fires inject only into the
boundary layer, so if the *amount* of small-fire smoke is available, its vertical distribution
can be assumed with some confidence.


**Data Availability**: The GEOS-Chem code used to generate this paper has already been
passed to the GEOS-Chem model support team and we currently plan to include it as an
option in the next public version of the model. The anticipated release date will be prior
to publication. The aircraft and surface data used in this paper is already publically

available.

**Appendices**: N/A

**Supplemental Information:** N/A

**Team List and Author Contributions:**

**Liye Zhu** led the majority of the analysis associated with this manuscript.

**Maria Val Martin** led the analysis of the MISR data and developed the monthly average
gridded climatology of plume heights for 2008.

**Arsineh Hecobian** provided the smoke designation associated with the ARCTAS aircraft
data.

**Luciana V. Gatti** led the aircraft measurements over the Amazon Basin.



**Ralph Kahn** helped develop the MISR plume-height algorithm, and led or mentored much of its application to wildfire smoke and volcanic plumes.

**Emily V. Fischer** led the conception of the work and the writing of this manuscript.

**Competing interests:** The authors declare that they have no conflict of interest.

**Disclaimer:** N/A

**Special issue statement:** N/A

**Acknowledgements** This work was supported by NASA Award Numbers NNX14AF14G and NNX14AN47G. PAN data from ARCTAS was provided by Greg Huey supported by NASA Award Number NNX08AR67G. Amazon vertical profile data were provided by L.V. Gatti supported by NERC (NE/F005806/1) and

FAPESP (08/58120-3). We thank Glenn Diskin for the use of the ARCTAS CO data. We thank Paul C. Novelli for the use of the CO data from NOAA ESRL Carbon Cycle Cooperative Global Air Sampling Network.

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




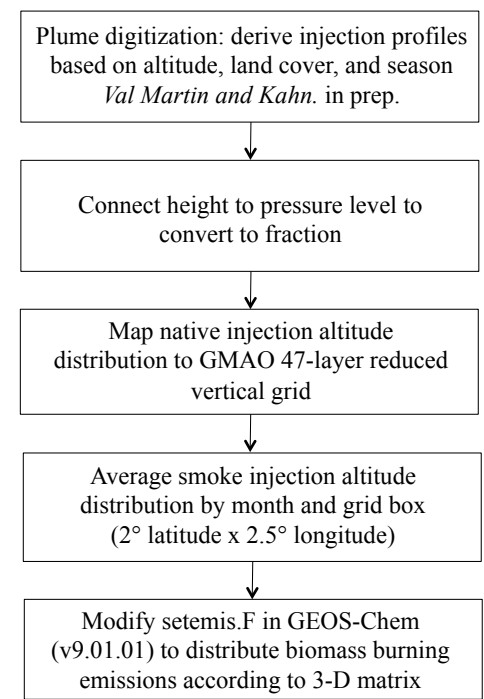

| Plume digitization: derive injection profiles based on altitude, land cover, and season *Val Martin and Kahn.* in prep. |
| Connect height to pressure level to convert to fraction |
| Map native injection altitude distribution to GMAO 47-layer reduced vertical grid |
| Average smoke injection altitude distribution by month and grid box (2° latitude x 2.5° longitude) |
| Modify setemis.F in GEOS-Chem (v9.01.01) to distribute biomass burning emissions according to 3-D matrix |


**Figure 1:** Overview of the implementation of an observationally based scheme to inject biomass burning emissions within GEOS-Chem.

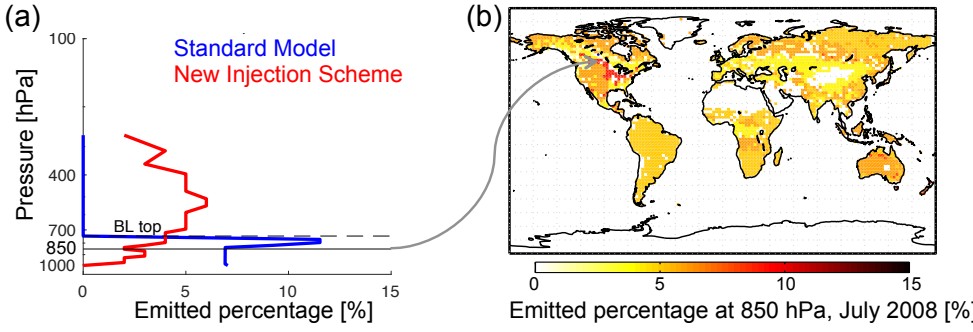


**Figure 2:** (a) Vertical profile of the percent of emissions in each model level for a sample location over boreal Canada (56°N, 105°W) from the public release version of GEOS-Chem (blue) and the new observationally-based injection scheme (red). The dashed line



indicates the averaged boundary layer top of this month. The solid black line is at 850
hPa, corresponding to the layer shown in (b). (b) Percent of total column biomass burning
emissions emitted into the 850 hPa layer in each model grid cell for July 2008.


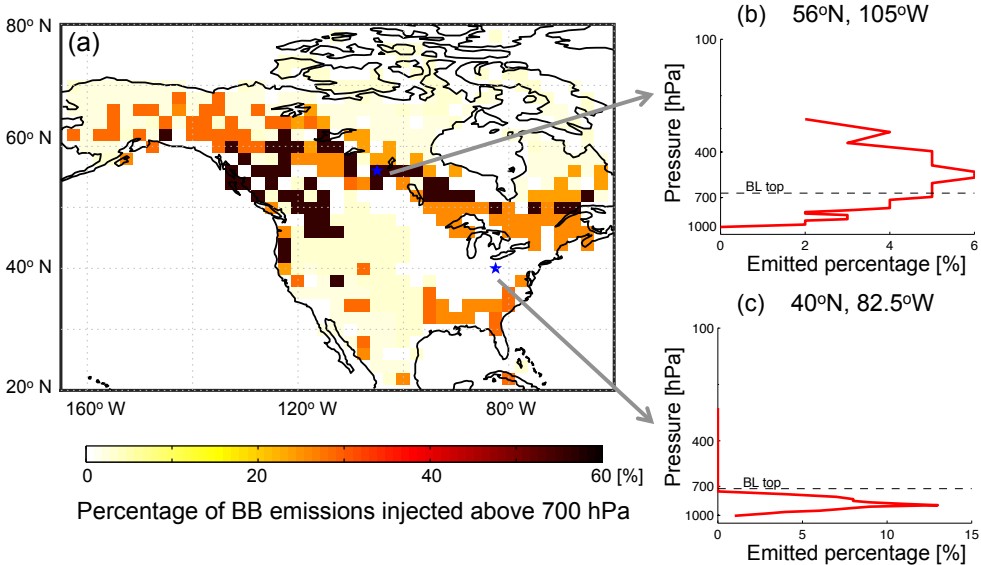

**Figure 3:** (a) Percentage of total column biomass burning emissions injected above 700
hPa over North America for July 2008, based on MISR observations. The two example
locations shown in (b) and (c) are marked as blue stars. (b) Vertical profile of the percent
of emissions in each model level over 56°N, 105°W. (c) Vertical profile of the percent of
emissions in each model level over 40°N, 82.5°W. The dashed line indicates the averaged
boundary layer top during this month.





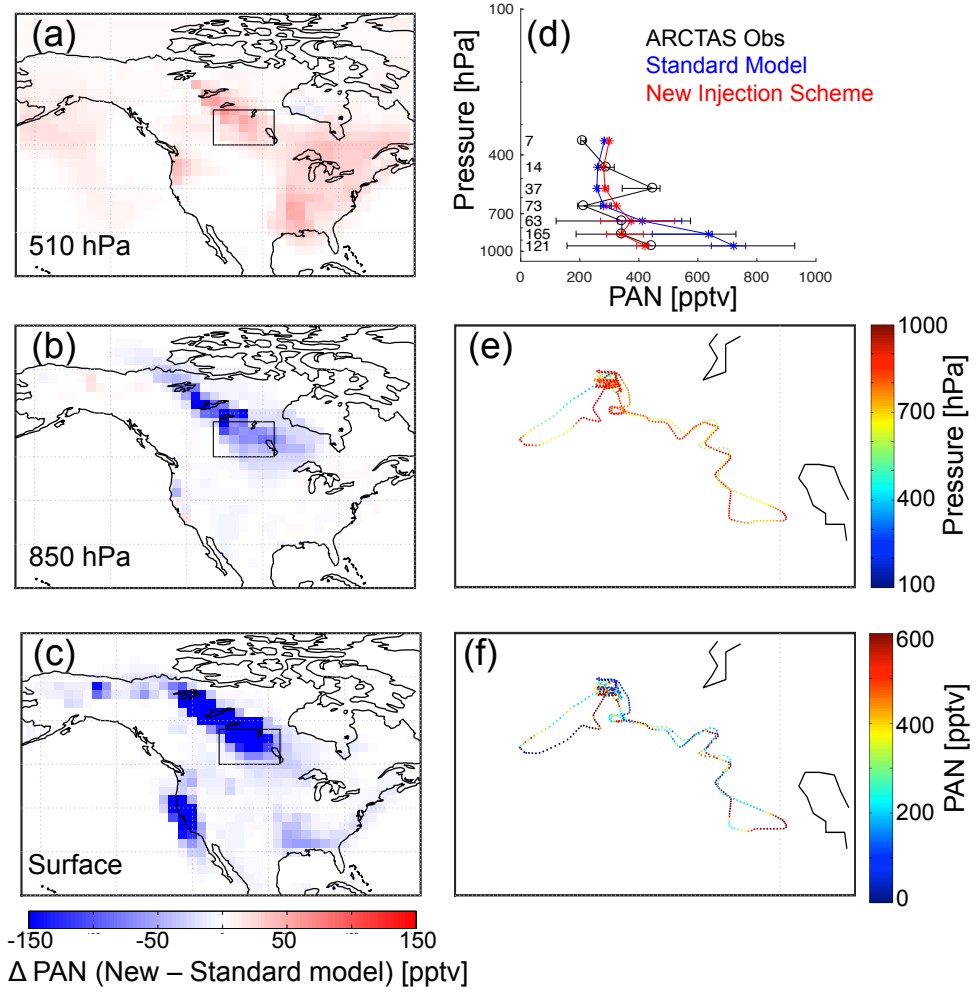


**Figure 4:** (a)-(c) Differences in simulated PAN mixing ratios between a GEOS-Chem
simulation with and without the new observationally based biomass burning injection
scheme over North America at three different levels: 510 hPa, 850 hPa, and surface ofor
July 1, 2008. (d) Median vertical profiles of ARCTAS PAN mixing ratios (black),
standard model (blue), and new injection scheme (red) on July 1, 2008. The whiskers
represent 25% and 75% percentiles of the data in the pressure bins. The numbers on the
left are the numbers of observations in different pressure bins. (e) ARCTAS in situ

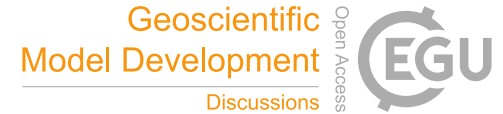



aircraft observations for July 1, 2008 colored by ambient pressure for the inset black box
in (a) - (c). (f) ARCTAS observations for July 1, 2008 colored by PAN mixing ratio.


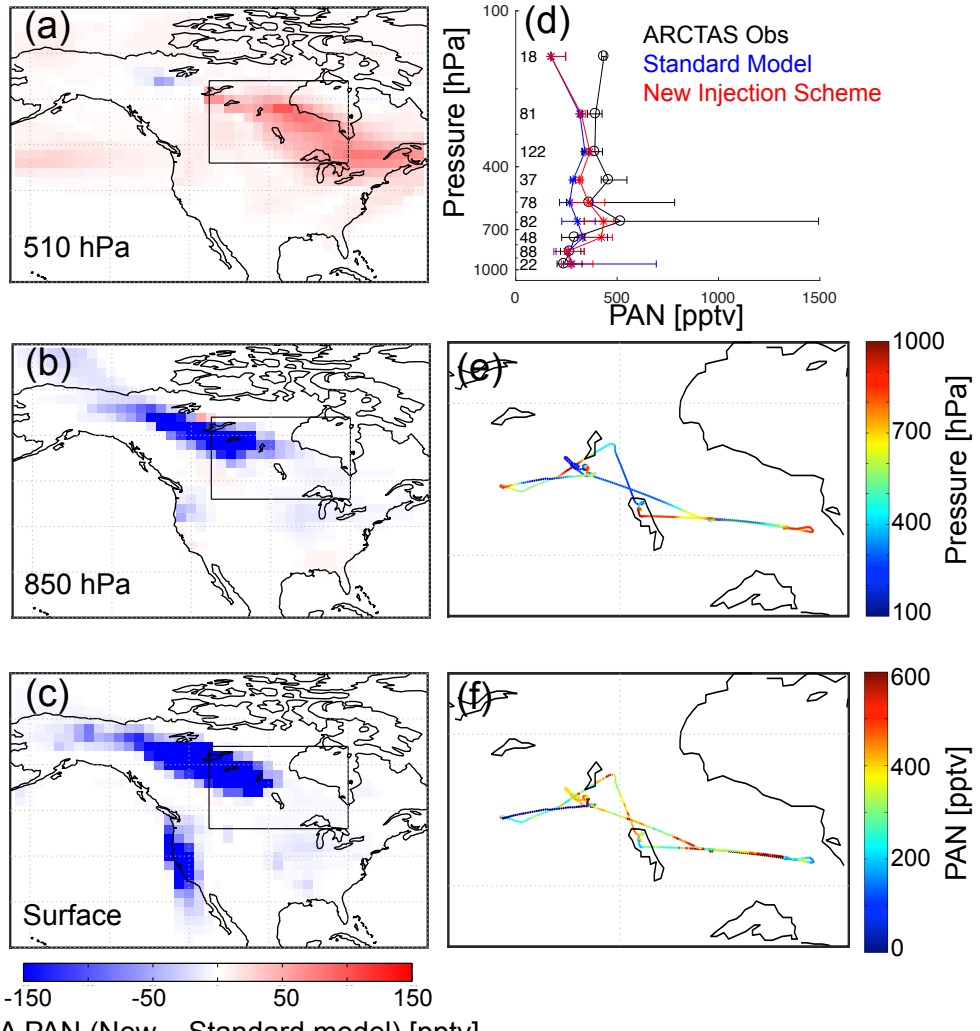

**Figure 5:** Same as Figure 4, but for July 4, 2008.




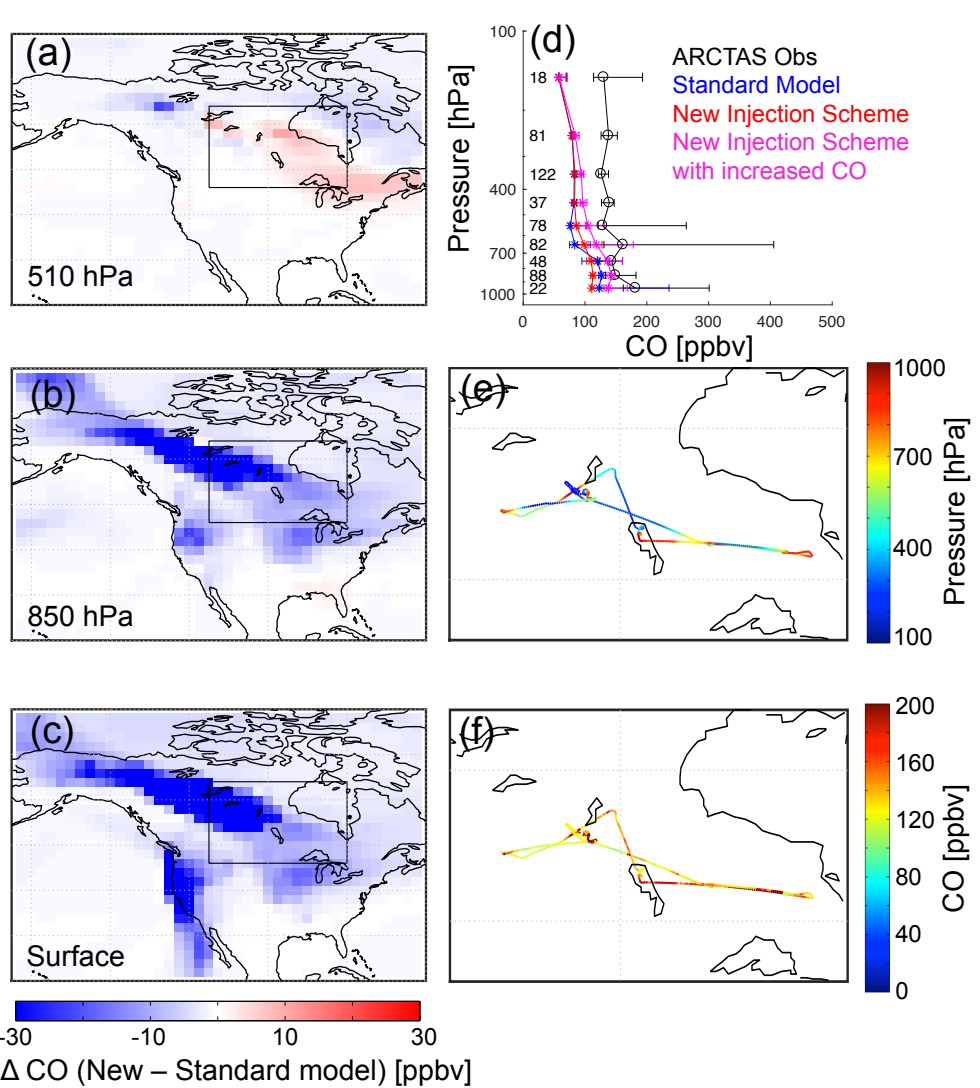

**Figure 6:** Same as in Figure 5, but for CO. The pink profiles are from a simulation that also increased the emissions of CO from boreal fires as described in Section 2.4.



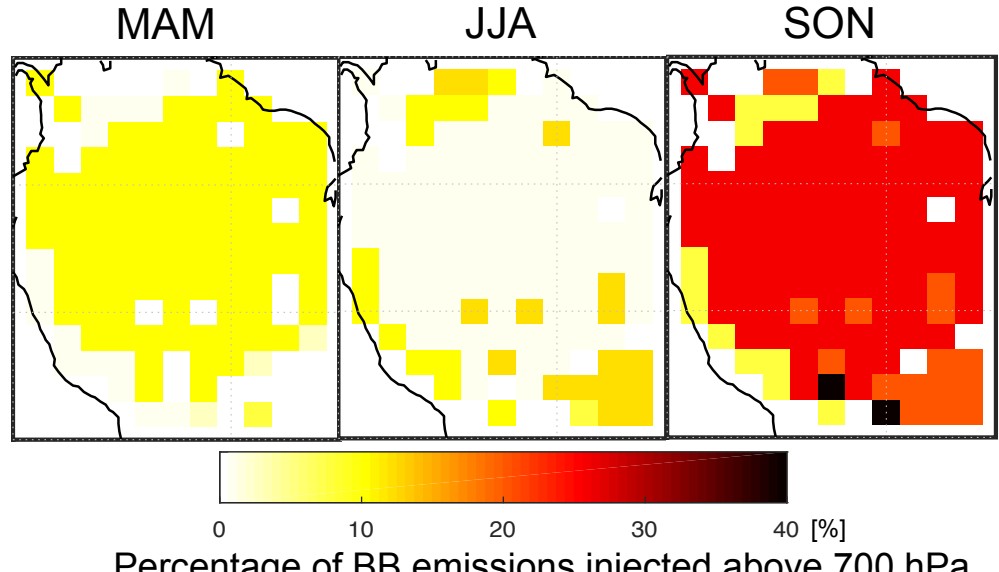

**Figure 7:** Percentage of total column biomass burning emissions injected above 700 hPa over the Amazon from March to November of 2010 and 2011, based on MISR plume-height analysis.





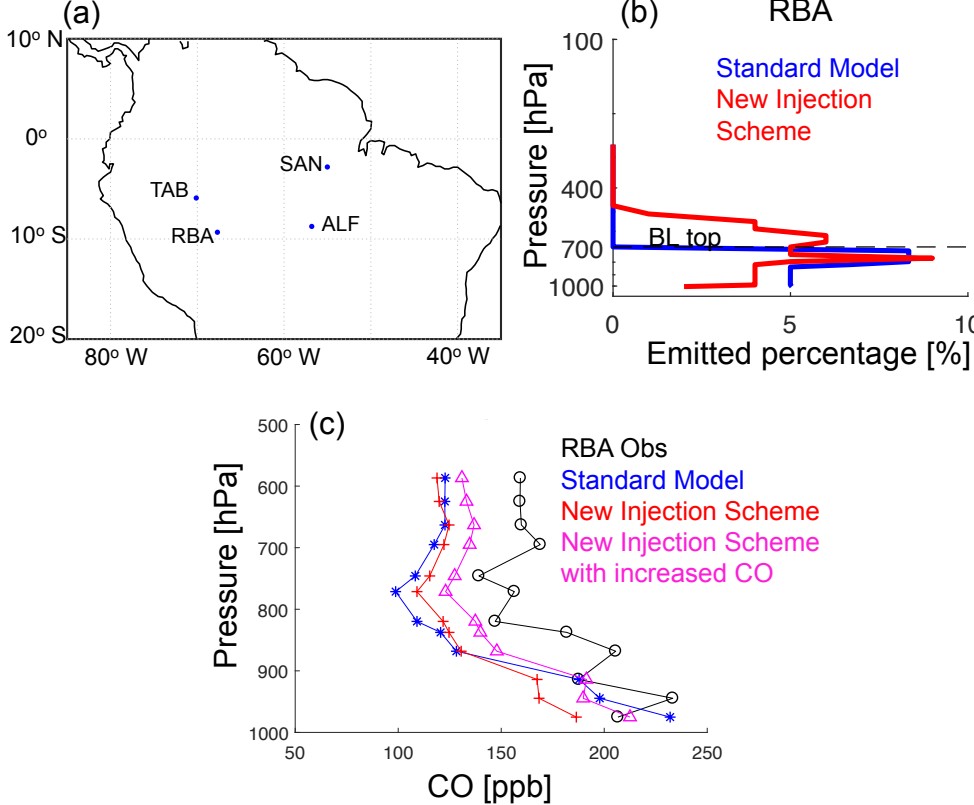

**Figure 8:** (a) Map of four measurement sites in the Amazon basin. (b) Vertical profile of

the percent of emissions in each model level at site RBA from the public release version

of GEOS-Chem (blue) and the new observationally-based injection scheme (red). The

dashed line indicates the averaged boundary layer top during this month. (c) Median

vertical profiles of CO mixing ratios observed at RBA (black), simulated with the

standard model (blue), simulated with the new injection scheme (red), and simulated with

the new injection scheme and with increased CO (Petrenko et al., 2017) in October of

2010 and 2011.



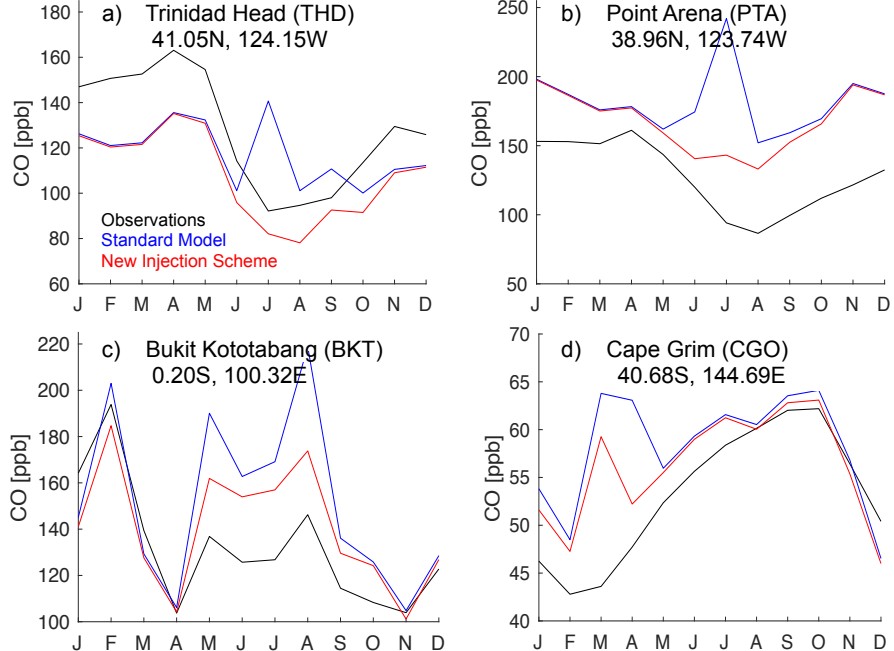

Figure 9: Observed and simulated monthly mean CO mixing ratios at select NOAA
ESRL Carbon Cycle Cooperative Global Air Sampling Network sites.
