# Peer review of "Development and implementation of a new biomass burning emissions injection height scheme (BBEIH v1.0) for the GEOS-Chem model (v9-01-01)"

_Geoscientific Model Development, 2018_

## Referee Comment (RC1) · Anonymous Referee #1 · 7 Jul 2018

This manuscript presents impacts of improved vertical distribution of biomass burning emissions in the GEOS Chem CTM model. The manuscript title is in somewhat misleading because it text is not self-contained regarding the description of the development of injection height parameterization. And actually, critical details about the developed parameterization is referred to an in preparation paper by Val Martin and Kahn. So, this paper should at least be published as a companion paper of the Val Martin and Kahn one. In particular, I believe a fair comparison between the standard and new injection schemes shown in section 3, would require to include the simulated profile of CO using the standard scheme with the increased CO emission.

[Figure]

Questions: Page 3 lines 124-128. The authors wrote: "MISR equator-crossing time during the day is about 10:30 AM, so the diurnal distribution of emissions is not sampled, and in particular, the mid-late afternoon, when wildfires tend to be most intense". The afternoon fires not only tend to produce more substantial emission rates but also they inject smoke in higher altitudes. How is this accounted for in the parameterization of the injection height?

Page 10 lines 318-325. The new injection scheme does not substantially improve the simulation of CO. It not only underestimate the CO amount in the entire atmospheric column but also produces a monotonic decrease from the surface to upper levels, not being capable of to simulate the enrichment layer present just above 700 hPa. The authors should comment on these features.

Page 11, section 3.2 Should be noted that for the Amazon basin 700 hPa is well above the boundary layer. So, the standard model also includes the lower part of the free troposphere.

Page 11, lines 372 . . . The new scheme injects a larger amount of CO above 700 hPa. However, no improvements are shown in the simulated CO profile above this height (figure 8C). The authors should try to explain this result.

Page 12, lines 378. . . I agree that increasing CO by 1.5 produces a better comparison with observations (figure 8C). But, are you not only rescaling? How about increasing by 2.0? Probably, the simulation will be even better. However, all the simulations present a monotonic decrease of CO, at least from the surface up to ∼850 hPa, while the observations show a more 'zig-zag' behavior with CO-enriched layers just above the surface and 900 hPa not present in the simulations. How to explain the model behavior? Too much vertical mixing in the model? Too coarse vertical resolution?

---

## Short Comment (SC2) · 9 Jul 2018

Dear Authors,

firstly, I asked the editorial team to change the paper type of your manuscript, as you are clearly describing a new development this should be a "Development and technical" paper and not an "Evaluation paper". A "Developement and technical paper" of course has to include an basic evaluation of the new development, but an evaluation paper does not really contain larger new developments. As a consequence, please provide the version number of GEOSChem model and a name (Acronym and version number of the newly developed scheme) in the title of the article. The title could look like:

"Development and implementation of a new biomass burning emissions injection height scheme (BBEIH vx.y) for the GEOSChem model (version z.g)"

Secondly, the paragraph you named "Data Availability" should be named "Code and data availability". As explained in
https://www.geoscientific-model-development.net/about/manuscript_types.html. GMD is encouraging authors to upload the program code of models (including relevant data sets) as supplement or make the code and data of the exact model version described in the paper accessible through a DOI (digital object identifier). In case your institution does not provide the possibility to make electronic data accessible through a DOI you may consider other providers (eg. zenodo.org of CERN) to create a DOI. Please note that in the code availability section you can still point the reader to how to obtain the newest version. If for some reason the code and/or data cannot be made available in this form (e.g. only via e-mail contact) the "Code Availability" section need to clearly state the reasons for why access is restricted (e.g. licensing reasons).
Especially, please note, that it is not enough, that the code will be available in the future. It must be available now and the exact version of the code published in this article needs to be made available.

Additionally, please note, that the exact code needs to be available to the Editor and the Referees for the review process.

Yours, Astrid Kerkweg

———————————————————

---

## Editor Comment (EC1) · J E Williams (Editor) · 11 Jul 2018

Dear Authors,

The anonymous comment which was posted in this discussion has been removed by the editorial office as it did not adhere to the instructions related to short-comments, namely:

Short comments (SCs) can be posted by any registered member of the scientific community (free online registration). Such comments are attributed, i.e. posted under the name of the commentator.

---

## Referee Comment (RC2) · Anonymous Referee #2 · 24 Jul 2018

This paper describes a MISR-based scheme for estimating the vertical distribution of biomass burning emissions in the GEOS-Chem model. Monthly gridded MISR injection heights from 2008 are used to develop the distribution for each month and each grid cell of a 2x2.5 degree GEOS-Chem grid. The paper demonstrates the impact that this new scheme has on GEOS-Chem predictions of CO and PAN over biomass burning regions, specifically Canada in July of 2008 (during the ARCTAS-B campaign) and over the Amazon in October of 2010 and 2011. They find that the new injection height scheme better matches observations of the vertical profile of PAN during ARCTAS-B and the vertical profiles of CO during ARCTAS-B and over the Amazon, as well as showing an improved match with NOAA ESRL surface observations of CO during 2008.

[Figure]

The need for better approaches to simulating the injection height of biomass burning emissions is clear, and this work to develop an empirically-based approach based on satellite observations is an important advance in the field. The data used to demonstrate the impacts of the new scheme seem reasonable and allow comparisons with published results from other versions of the GEOS-Chem model. However, the manuscript is occasionally unclear and lacks key details in describing the new scheme, and the order of the discussion is sometimes repetitive and confusing. Thus, I recommend minor revisions to address the minor concerns and typos below.

Minor Concerns

L116-117, L148-150: I am not clear on how the MISR injection height data are converted into emitted percentages of biomass burning emissions on the GMAO grid at 2x2.5 degree resolution. Are all MISR heights in a given month and 2x2.5 degree grid box averaged together, weighted by their relative emissions? Is the land cover used to define that weighting? Since the final product is monthly on a 2x2.5 grid, what do the words "region" and "season" in L116-117 refer to? A little more description and possibly some equations would help to make the data processing clear.

L125-127: You mention the morning observation time of MISR as a limitation several times. What kind of errors do you expect this limitation would have on the model results? Have you considered any potential correction for this limitation, such as applying a normalized diurnal cycle to the MISR observations?

L130-131: My understanding is that even if the fires are smaller than a pixel, they can be detected if they have a sufficient impact on the brightness temperature of the pixel, so while some fires are too small to detect, not all fires smaller than a MODIS pixel are missed. Is that correct? If so, this sentence needs to be revised.

L136-137: Can you make a case that the Randerson et al. (2012) approach for small fires is likely to be accurate? Why did you pick this approach?

L139: There is a lot of material left for the Val Martin and Kahn (in prep) paper – I'd suggest adding a little more detail from that reference here as it will be hard for future readers to track down the other paper without a reference.

L153: You might want to mention here why you chose this version – I think it's because it is what was used in Fischer et al. (2014), as stated on L175, but it should be made clear here at the first mention of the version.

L168-170: The phrase "As discussed in the preceding and following sections" is not very helpful in finding where the discussion is. Perhaps replace it with "As discussed in Section 2.2, since MODIS and MISR may not detect many small fires,..."

L181-182: You might want to clarify what you mean by "best" here – most chemically detailed, most accurate, most studied – and provide some brief evidence why this version of GEOS-Chem is the best for PAN in that sense.

L184-185: I don't understand how this sentence on standard input file settings and benchmark runs links with the previous sentence on the problems with evaluating monthly-average emissions with specific case studies. Do you mean that since most users will run using monthly-average biomass burning emissions, that's why you used that approach in this paper?

L197-199: Rather than refer back to a paper we can't read yet, I'd suggest referring back to Section 2.2, where you discuss this issue in a little more detail.

L200-220: The ordering of the sentences in this paragraph was very confusing to me. I think the point is to say that your approach in this paper follows Fischer et al. (2014) with exceptions, and then to list those exceptions. However, you start by saying what you removed from Fischer et al. (2014), then explain why Fischer et al. (2014) is your "standard" model, then you introduce a new CO emission factor approach, then state the horizontal resolution of the model. I think this could be made much clearer by rearranging the content.

L239-241: Is there some reason it is important to mention that everything was synchronized to UTC? Wouldn't any other time zone work as well so long as you were consistent?

L252: How were the CO measurements over the Amazon performed?

L254-256: I'm not sure what this sentence means – how else would I get a vertical profile except by measuring at specific altitudes? Are you contrasting this with remote sensing approaches?

L257-266: Since you don't use the BARCA data later in the paper, I'd cut this part of the paragraph. However, you do use the NOAA ESRL data later (Section 3.3), so a discussion of those data should be included here.

L318: Why are you not showing the CO results for the 1 July flight as well?

L336: I think you need to explain why you did not pursue the daily or 3-hour emission approaches, and/or why you think the approach using data from Petrenko et al. (2017) is better.

L341-342: The match between 900 and 800 hPa looks like a coincidence to me, as the model profile is highest there but it is a local minimum in the observed profile. Can you argue that I should have more confidence in the match in that region?

L343-350: This "limitations and future work" paragraph would probably fit better in Section 4.

L358-359: For all of these "in prep" references, it would be good to either add more detail to this paper or provide a better reference, such as to a conference presentation if one exists.

L409: But there is a large negative change in Northwest Canadian Figure 6a, which does not seem consistent with you OH-based explanation. What is the cause of that feature?

L425-430: Rather than referring to "most" stations with no change and "some" with decreases, can you be quantitative? How many sites had no change (and does this mean a change of less than 1 ppb? 10 ppb?), and how many had decreases? Did any have increases?

L467-469: You don't present any comparisons to satellite observations in this paper, so this doesn't really belong in your summary.

Typos and Style Suggestions

L98: Replace "this trace species" with "PAN" to be clear.

L134: I'd suggest that "account for" is closer to what you mean than "acknowledge" here.

L137: Extra "s" after "biome"

L330: Extra comma after "specific"

L352: I don't think you need the word "above" as you mention the section number.

L371-372: You discuss the results in Figure 8c before introducing the figure. I'd cut the previous sentence to "...emissions pushed higher in the atmosphere."

L394: I'm not sure "understand" is the right word here. Maybe "simulate"?

L398-399: Instead of referring to the "example in Figure 6", I'd suggest describing it as "the 4 July smoke plume from ARCTAS (Fig. 6)".

L412: Word "typically" is redundant with "Typical" at the beginning of the sentence.

L424: Extra space after "Network"

L463: I'd suggest changing this to "provided access to CO profiles that could be used for model-measurement comparison"

L464: "do not appear", instead of "to not appear"

---

## Editor Comment (EC2) · J E Williams (Editor) · 1 Aug 2018

Dear Authors,

The anonymous comment which was posted in this discussion has been removed by the editorial office as it did not adhere to the instructions related to short-comments, namely:

A referee of a manuscript should judge objectively the quality of the manuscript and respect the intellectual independence of the authors. In no case is personal criticism appropriate.

There were an number of unsubstantiated accusations in the comment which cannot be corroborated and seemed to originate from some potential conflict outside the remit of the GMD platform.

Jason Williams.

**[GMDD](https://doi.org)**

---

## Author Comment (AC1) · 21 Sep 2018

Please see the Supplement

Please also note the supplement to this comment:
https://www.geosci-model-dev-discuss.net/gmd-2018-93/gmd-2018-93-AC1-supplement.zip

---

## Author Response (AR1)

We appreciate the comments from the reviewers. We have carefully considered all the comments and revised our manuscript. We hope that the revisions improve the paper. The following responses address all the reviews' comments in a point-by-point fashion.

*Anonymous Referee #1*

*This manuscript presents impacts of improved vertical distribution of biomass burning emissions in the GEOS Chem CTM model. The manuscript title is in somewhat misleading because it text is not self-contained regarding the description of the development of injection height parameterization. And actually, critical details about the developed parameterization is referred to an in preparation paper by Val Martin and Kahn. So, this paper should at least be published as a companion paper of the Val Martin and Kahn one.*

The "development" in the title actually means the processes of adjusting the injection height fractions for the GEOS-Chem model and the coding processes of applying the new injection height scheme to the GEOS-Chem model. "Development" does not refer to the development of the injection height parameterization. We apologize for the misunderstanding. However, according to another reviewer's comments, we modified the title as "Development and implementation of a new biomass burning emissions injection height scheme (BBEIH v1.0) for the GEOS-Chem model (v9-01-01)". The parameterization paper by Val Martin and Kahn has been submitted to *Remote Sensing*. Thus, we replaced all the citations as "Maria Val Martin and Ralph A Kahn (2018), A Global Climatology of Wildfire Smoke Injection Height Derived from Space-based Multi-angle Imaging, submitted to Remote Sensing (MISR Special Issue), manuscript ID remotesensing-359296." We also upload the manuscript of this paper along with our responses.

*In particular, I believe a fair comparison between the standard and new injection schemes shown in section 3, would require to include the simulated profile of CO using the standard scheme with the increased CO emission.*

We thank the reviewer's suggestion. We have now included results from a simulated profile of CO using the standard scheme with the increased CO emissions in Figure 6, green lines. We have also added the following description in the main text.

"We also include results from a simulation from the standard model with increased CO emissions (green line in Figure 7d). The green line indicates that this model configuration substantially increases the CO concentrations within the boundary layer as expected. Comparing this simulation (green line in Figure 7d) to the simulation incorporating both new injection scheme and increased CO emissions (pink line in Figure 7d), shows the impact of both changes. By comparison, CO is higher at levels above the boundary layer and slightly lower in the boundary layer.

*Questions: Page 3 lines 124-128. The authors wrote: "MISR equator-crossing time during the day is about 10:30 AM, so the diurnal distribution of emissions is not sampled, and in particular, the mid-late afternoon, when wildfires tend to be most intense". The*

*afternoon fires not only tend to produce more substantial emission rates but also they inject smoke in higher altitudes. How is this accounted for in the parameterization of the injection height?*

The parameterization does not account for the afternoon fire peak and that limitation is discussed in detail in the parameterization paper (Val Martin and Kahn, 2018). The Val Martin and Kahn paper suggests that "at least a qualitative assessment of the diurnal representativeness of the MISR plume-height record might be made by comparing the FRP from Terra MODIS with corresponding values from satellites in other polar orbits, such as the MODIS instrument on NASA's Aqua satellite, and possibly geostationary FRP detectors. Such extensions would be worth exploring, but are beyond the scope of the current study." We do not repeat this information exactly in the current manuscript. Instead, we have added the following sentences to explain the limitation.

"In order to evaluate the impact of the afternoon peaks on the parameterization, a qualitative assessment of the diurnal representativeness of the MISR plume-height record is required, as well as the corresponding FRP data from other satellite instruments. Limitations of the parameterization are further discussed in Val Martin and Kahn (2018, submitted to Remote Sensing), and would be worth exploring in the future."

*Page 10 lines 318-325. The new injection scheme does not substantially improve the simulation of CO. It not only underestimate the CO amount in the entire atmospheric column but also produces a monotonic decrease from the surface to upper levels, not being capable of to simulate the enrichment layer present just above 700 hPa. The authors should comment on these features.*

The paragraph already pointed out deficiencies in the CO simulation at this location; however, we have added the additional specific comments suggested by the reviewer. This now reads:

"However, both the standard model and the new injection scheme underestimate CO significantly compared to ARCTAS observations. Both model versions continue to produce a monotonic decrease in CO from the surface to upper levels, and although the new injection scheme increases CO just above 700 hPa, it is not able to simulate the enrichment layer that appears present in the observations. The mean CO underestimate shown in Figure 7d is 15%-56%. The model does not appear to have such a low bias for the 1 July case (Figure 5), but there are very few samples at higher altitudes in this flight."

*Page 11, section 3.2 Should be noted that for the Amazon basin 700 hPa is well above the boundary layer. So, the standard model also includes the lower part of the free troposphere.*

The reviewer is correct here. Maria Val Martin's student examined PBL heights over the Amazon with MERRA-2. For October-November, she estimates the altitudes are about 1200-1500 m above ground level, which corresponds to 880-840 hPa. These values are

monthly averages from morning/early afternoon (11-13 local time) PBL heights over Tropical, Savanna and Grasslands across the Amazon (Gonzalez-Alonso et al, submitted to ACP). We have added this information to Section 3.2.

*Page 11, lines 372 . . . The new scheme injects a larger amount of CO above 700 hPa. However, no improvements are shown in the simulated CO profile above this height (figure 8C). The authors should try to explain this result.*

The new scheme indeed injects about 25% of biomass burning CO above 700 hPa. However, as described in Section 3.3, once emitted higher in the atmosphere, this fraction of the biomass burning CO would react more quickly with OH than in the standard simulations. We have included more text to clarify this near the discussion of Figure 8c in Section 3.3.

"Thus when a fraction of the CO emissions are immediately moved out of the boundary layer, this fraction reacts more quickly with OH than in the standard simulation. The same issue applies throughout the atmosphere, and can be visualized for the Amazon region in Figure 8c. The CO mixing ratio decreases with altitude above 650 hPa at a faster rate in the simulation with the new injection scheme than in the standard model. This effect is not local to a given fire, but reflects the cumulative impact of changing the emission altitude for a substantial quantity of CO emissions."

*Page 12, lines 378. . . I agree that increasing CO by 1.5 produces a better comparison with observations (figure 8C). But, are you not only rescaling? How about increasing by 2.0? Probably, the simulation will be even better. However, all the simulations present a monotonic decrease of CO, at least from the surface up to ∼850 hPa, while the observations show a more 'zig-zag' behavior with CO-enriched layers just above the surface and 900 hPa not present in the simulations. How to explain the model behavior? Too much vertical mixing in the model? Too coarse vertical resolution?*

We didn't simply increase CO by a factor of 1.5 globally. The magnitudes of scaling factors used for the CO emission factor are based on land use types and the analysis in Petrenko et al. (2017). Thus, it is not appropriate to adjust the CO emission with an arbitrary factor even if it may match the observations slightly better.

The reviewer is right. The model is not able to catch the enriched layers just above the boundary layer. However, there are many possible reasons for this. 1) The emission injection fraction is based on a monthly averaged dataset, and then it is averaged for each model level. The observations reflect a specific time and location, which could reflect a particularly intense fire with a higher than average injection altitude profile. 2) Though clearly the emission inventory is capturing some of the fires, it is also possible that the emission inventory is missing a particular fire in the region. This could occur for a multitude of reasons.

*Anonymous Referee #2*

*This paper describes a MISR-based scheme for estimating the vertical distribution of biomass burning emissions in the GEOS-Chem model. Monthly gridded MISR injection heights from 2008 are used to develop the distribution for each month and each grid cell of a 2x2.5 degree GEOS-Chem grid. The paper demonstrates the impact that this new scheme has on GEOS-Chem predictions of CO and PAN over biomass burning regions, specifically Canada in July of 2008 (during the ARCTAS-B campaign) and over the Amazon in October of 2010 and 2011. They find that the new injection height scheme better matches observations of the vertical profile of PAN during ARCTAS- B and the vertical profiles of CO during ARCTAS-B and over the Amazon, as well as showing an improved match with NOAA ESRL surface observations of CO during 2008.*

*The need for better approaches to simulating the injection height of biomass burning emissions is clear, and this work to develop an empirically-based approach based on satellite observations is an important advance in the field. The data used to demonstrate the impacts of the new scheme seem reasonable and allow comparisons with published results from other versions of the GEOS-Chem model. However, the manuscript is occasionally unclear and lacks key details in describing the new scheme, and the order of the discussion is sometimes repetitive and confusing. Thus, I recommend minor revisions to address the minor concerns and typos below.*

We thank the reviewer's valuable comments. We have addressed all the comments below.

*Minor Concerns*

*L116-117, L148-150: I am not clear on how the MISR injection height data are converted into emitted percentages of biomass burning emissions on the GMAO grid at 2x2.5 degree resolution. Are all MISR heights in a given month and 2x2.5 degree grid box averaged together, weighted by their relative emissions? Is the land cover used to define that weighting? Since the final product is monthly on a 2x2.5 grid, what do the words "region" and "season" in L116-117 refer to? A little more description and possibly some equations would help to make the data processing clear.*

We have added the following information to Section 2.2.

"Briefly, MISR-based injection heights are given by altitude (250 m, from 0 to 8 km above ground level), land cover type, season and region. Land cover classifications are based on MODIS Level 3 land cover product MOD12Q1 (Friedl et al., 2010). There are twelve classifications used here: Evergreen Needle Leaf Forest, Evergreen BroadLeaf Forest, Deciduous Needle Leaf Forest, Deciduous BroadLeaf Forest, Mixed Forest, Closed Shrub, Open Shrub, Woody Savanna, Savanna, Grassland, Wetland and Cropland. We defined seasons as spring (MAM), summer (JJA), fall (SON) and winter (DJF), and considered 8 main fire regions (North America, South America, Africa, Europe, Boreal Eurasia, South Asia and Australia).

To convert the MISR-based vertical distribution of smoke injection height, Val Martin and Kahn (2018) first transformed the MISR vertical distribution percentages from 0 to 8 km at 250 m bins into the GEOS-Chem 47 level vertical grid (0.058, 0.189, 0.32, 0.454,

0.589, 0.726, 0.864, 1.004 km, etc). Second, they determined the largest land cover type coverage in each GEOS-Chem grid. For that, they re-gridded their land cover type map from 0.005° × 0.005° to 2° × 2.5° degree resolution assigning the highest ranked land cover type to each 2° × 2.5° grid. Finally, they applied the re-gridded vertical distribution of smoke percentages to each 2° × 2.5° degree grid depending on the defined land cover type and region."

*L125-127: You mention the morning observation time of MISR as a limitation several times. What kind of errors do you expect this limitation would have on the model results? Have you considered any potential correction for this limitation, such as applying a normalized diurnal cycle to the MISR observations?*

The morning observation time of MISR is a limitation because it may miss the peak afternoon fires at some locations. Fire activity usually peaks in the afternoon or early evening, and at these times, fires have larger FRP and higher injection heights. So it is likely that the MISR observations of smoke plumes are biased low and the emitted fraction in upper levels is likely a lower bound. The parameterization paper- Val Martin and Kahn (2018, submitted to Remote Sensing), also describes this limitation and considers options to correct for this known bias by qualitatively assessing the diurnal representativeness of the MISR plume-height as compared to FRP based on the observations from Terra MODIS or other satellite instruments. Our study is the first study to apply the new injection scheme globally, and a series of improvements would be expected in the future work. We think it is far beyond the scope of this current paper to apply corrections at this point. This step forward may be possible after the upcoming series of fire-focused field campaigns however.

*L130-131: My understanding is that even if the fires are smaller than a pixel, they can be detected if they have a sufficient impact on the brightness temperature of the pixel, so while some fires are too small to detect, not all fires smaller than a MODIS pixel are missed. Is that correct? If so, this sentence needs to be revised.*

The reviewer is correct that fires smaller than a pixel can be detected if the brightness temperature is high. However, many small fires have an emissivity at 4 microns that are very low and they can be missed by MODIS (Kahn, et al 2007). The other limitation is that small fires may sometimes be overseen by the MINX digitizer users, and/or can be digitized with low quality as they have low stereo-height retrieval densities. We have revised this section to include these additional details. This now reads:

"Several factors contribute to this limitation. MODIS thermal anomalies are used to identify fire locations, some fires are smaller than MODIS pixels, others can be obscured by the tree canopy or overlying smoke, and fires for which the emissivity at 4 microns is low (e.g., smoldering fires), are sometimes missed (Kahn et al., 2008). These issues also affect satellite-based smoke emissions inventories such the one used here (see Section 2.4). The other limitation is that small fires may sometimes be overseen by the MINX digitizer users, and/or can be digitized with low quality as they have low stereo-height

retrieval densities."

*L136-137: Can you make a case that the Randerson et al. (2012) approach for small fires is likely to be accurate? Why did you pick this approach?*

There is ample evidence suggesting that the MISR will miss some small fires and the majority of small fires inject smoke only into the boundary layer. Thus, to account for small fires that are typically under-detected by MISR, we apply a correction to the lowest level of our vertical profiles (0-250 m). Below we show a figure of the vertical distribution of the percentage of smoke calculated without correction (black) and adjusted with the GFED4s fraction to account for small fires (green), over cropland fire in Europe and forest fires in North America. This figure is the supplemental Figure S6 in the parameterization paper (Val Martin and Kahn, 2018, submitted to Remote Sensing). For cropland fires over Europe during the summertime, we apply a correction of 30%, and the percentage of smoke injected in the lowest level is increased from 11.6 to 15.2%. The fraction of small fires over forests in North America is smaller (13%) and thus the increase is only from 5.1 to 5.7%. We also included a table from the Val Martin and Kahn parameterization paper that shows the percentages of corrections to the lowest level of vertical distribution. We view the correction approach used in Randerson et al. (2012) as a conservative approach because it places more smoke at lower altitudes and the relative magnitude of the adjustment makes sense because cropland and grassland fires have the largest adjustments as fires over these land cover types are typically smaller than forest fires.

[Figure]

**Figure S6.** Vertical distribution of percentage of smoke calculated with the original+AOD-filled retrievals (black) and original+AOD-filled retrievals adjusted with the GFED4s fraction to account for small fires (green), over cropland fire in Europe and forest fires in North America.

**Table S2.** Percentages (minimum and maximum) applied to the vertical distribution lowest level to account for small fires under detected by MISR.

|  | Forest | Savanna | Grassland | Cropland |
|---|---|---|---|---|
| North America | 13–30 | 15–36 | 10–30 | 31–44 |
| South America | 29–43 | 22–33 | 20–34 | 24–38 |
| Africa | 35–50 | 16–44 | 11–25 | 22–31 |
| Europa | 36–50 | 28–47 | 27–49 | 31–48 |
| Boreal Eurasia | 23–50 | 8–46 | 28–39 | 36–46 |
| South Asia | 34–45 | 12–45 | 10–30 | 38–45 |
| Australia | 27–50 | 11–26 | 7–42 | 36–46 |

*L139: There is a lot of material left for the Val Martin and Kahn (in prep) paper – I'd suggest adding a little more detail from that reference here as it will be hard for future readers to track down the other paper without a reference.*

The "Val Martin and Kahn (in prep) paper" has been submitted to *Remote Sensing*, and we also uploaded a copy of the manuscript along with our responses. We have also added substantially more details in Section 2.2 to help readers understand what is contained in Val Martin and Kahn without having to read that paper.

*L153: You might want to mention here why you chose this version – I think it's because it is what was used in Fischer et al. (2014), as stated on L175, but it should be made clear here at the first mention of the version.*

We have reorganized the order of describing the model configuration and implementation. So the reason for choosing this version has been described at the first time mentioned. See line 190-225.

*L168-170: The phrase "As discussed in the preceding and following sections" is not very helpful in finding where the discussion is. Perhaps replace it with "As discussed in Section 2.2, since MODIS and MISR may not detect many small fires, . . ."*

Since we reorganized the contents in Section 2.3 and 2.4, we deleted this particular sentence, but there is a direct reference to Section 2.2. now.

"Given the combined limitations in the MISR analysis (Section 2.2) and the GFED…"

*L181-182: You might want to clarify what you mean by "best" here – most chemically detailed, most accurate, most studied – and provide some brief evidence why this version of GEOS-Chem is the best for PAN in that sense.*

We added a few more sentence to clarify this sentence. That sentence was added in the initial editorial review, and now it does seem out of place. See line 190-195.

"The version of GEOS-Chem that we chose for developing and implementing the improved injection height scheme includes a number of code updates focused specifically on providing a better representation of PAN chemistry. It includes a more detailed chemical mechanisms related to PAN and a larger suite of precursor NMVOCs emissions. This model version has also been compared to a large suite of aircraft

observations."

*L184-185: I don't understand how this sentence on standard input file settings and benchmark runs links with the previous sentence on the problems with evaluating monthly-average emissions with specific case studies. Do you mean that since most users will run using monthly-average biomass burning emissions, that's why you used that approach in this paper?*

We mean that we are using the standard input file setting for the biomass burning emissions, which is also used by most users and benchmarks. However, we also want to point out the possibility that using monthly-average biomass burning emissions may lead biases for specific case studies. We apologize for the misunderstanding. We have modified the sentence as follow:

"we have used the standard input file settings used in GEOS-Chem. However, we note that choosing a monthly-averaged emission dataset can create biases for specific case studies of biomass burning"

*L197-199: Rather than refer back to a paper we can't read yet, I'd suggest referring back to Section 2.2, where you discuss this issue in a little more detail.*

Thanks for the suggestion. Now we refer back to Section 2.2.

*L200-220: The ordering of the sentences in this paragraph was very confusing to me. I think the point is to say that your approach in this paper follows Fischer et al. (2014) with exceptions, and then to list those exceptions. However, you start by saying what you removed from Fischer et al. (2014), then explain why Fischer et al. (2014) is your "standard" model, then you introduce a new CO emission factor approach, then state the horizontal resolution of the model. I think this could be made much clearer by rearranging the content.*

Thanks for the suggestions. We rearranged the content as suggested.

"The model experiments in Fischer et al. (2014) were among main motivations for the current paper. Thus, our model configurations are mainly based on the configuration used in this earlier study. However, the current work is focused on understanding potential changes in model performance following the inclusion of the new MISR-based injection height scheme. To keep this focus, there are two differences between the model configuration in Fischer et al. (2014) and our "standard model." 1) We adjust the biomass burning emissions used in Fischer et al. (2014). Specifically, we remove the increased biomass burning emissions for northern Asia, originally applied for 2008 in Fischer et al. (2014). These were applied in Fischer et al. (2014) because Kaiser et al. (2012) and Yu et al. (personal communication) found that GFEDv3 underestimates

fire emissions at boreal latitudes. 2) We also remove the injection partitioning assumption applied in Fischer et al. (2014), which emitted 35% of total biomass burning emissions above the boundary layer to test the sensitivity of PAN to this choice. Fischer et al. (2014) found this to improve the PAN simulation, but this is a much coarser approach than what has been done here.

In the following text and figures, we refer to the version of model with the two changes notes above as the "standard model" because the injection of biomass burning is treated as in the public release benchmarked version of GEOS-Chem. We refer to the observationally based injection scheme as the "new injection scheme." As is discussed later, we then apply different scaling factors for fire emissions following Petrenko et al. (2017) (see below) to the "new injection scheme." We refer to this final model configuration in our figures as the "new injection scheme with increased CO." "

*L239-241: Is there some reason it is important to mention that everything was synchronized to UTC? Wouldn't any other time zone work as well so long as you were consistent?*

We agree this is an unnecessary, and thus potentially confusing detail. We have removed this sentence.

*L252: How were the CO measurements over the Amazon performed?*

We added the following sentence to describe this briefly. "As described in Gatti et al. (2014), samples were collected using a small aircraft. Air samples were collected in flasks that were analyzed using a replica of the NOAA Earth System Research Laboratory (ESRL) trace gas analysis system."

*L254-256: I'm not sure what this sentence means – how else would I get a vertical profile except by measuring at specific altitudes? Are you contrasting this with remote sensing approaches?*

We agree this was confusing, but it should be much clearer now that we have added more information about how the samples were collected in response to the last comment. The series of sentences is now:

"As described in Gatti et al. (2014), samples were collected using a small aircraft. Air samples were collected in flasks that were analyzed using a replica of the NOAA Earth System Research Laboratory (ESRL) trace gas analysis system. The measurements were taken at specific altitude levels on each flight day. Up to six or eight observations are available at each individual altitude level for each month (4 sites with 2 vertical

profiles).”

*L257-266: Since you don't use the BARCA data later in the paper, I'd cut this part of the paragraph. However, you do use the NOAA ESRL data later (Section 3.3), so a discussion of those data should be included here.*

We indeed use the BARCA data for comparison. We didn't show any plots, as there is no difference between two model versions. However, we think it is still necessary to mention the comparison results with BARCA data so that readers realize this is not an oversight. We would like to keep this part.

The reviewer is right that we use NOAA ESRL data later. We have added a description of NOAA ESRL data to this section.

**"2.5.3 Surface Observations**

Leung et al. (2007) showed that the choice of injection height for boreal fire emissions impacts the simulation of surface CO mixing ratios in the Northern Hemisphere. They compared GEOS-Chem simulated anomalies in CO mixing ratios with surface measurements from the NOAA ESRL Global Monitoring Division (GMD), Carbon Cycle Cooperative Air Sampling Network (Novelli et al., 2003). Therefore, we also performed a comparison with monthly mean observations from 18 sites that may have been impacted by fires during 2008. In most locations (16 of 21) where we conducted comparisons, the model with the MISR-based injection height did not produce notably different surface monthly mean CO mixing ratios (i.e. changes are less than 1 ppb). However, there are four stations where the updated model produces substantially lower monthly mean surface CO mixing ratios than the standard model, and this change produces a better simulation of CO at these locations. We present these results in Section 3.3.”

*L318: Why are you not showing the CO results for the 1 July flight as well?*

We had made this choice for a few reasons. 1) The original Figure 6 (4 July flight CO results) demonstrates a more common problem with GEOS-Chem, i.e. the low CO bias, and there are already 9 figures. 2) The 1 July case has relatively few aircraft samples above 700 hPa. We actually debated not showing the PAN data either for this day. 3) Moving the emissions higher in the atmosphere degrades the performance of the model in the boundary layer compared to this flight. We are unsure what this signifies. However, so this is not confusing, we have added a figure showing the CO results for the 1 July flight to the manuscript (now Figure 5). We have added several sentences to reference this additional figure, and we have adjusted the figure numbering for subsequent figures.

[Figure]

*L336: I think you need to explain why you did not pursue the daily or 3-hour emission approaches, and/or why you think the approach using data from Petrenko et al. (2017) is better.*

We know that the emission inventory has a low bias, and thus, switching to daily or 3-hour emission approaches would not likely change the situation. In the end we decided to use the default settings for GEOS-Chem to show the expected impact. Our implementation is likely to eventually be used with a variety of different emission inventories. In addition, the biomass burning emission adjustment approach for GFEDv3 used in Petrenko et al. (2017) can successfully reproduce the satellite observed AOD data from MODIS. So we went with this approach. We have modified the text as follows.

"However, both the daily or 3-hourly emissions inventories in our case are still likely to be an underestimate of the true emissions. Thus, we did not pursue these options.

However, to simply show the impact of changing the emission factors, we include an additional simulation (pink line in Figure 7d) with both the updated injection scheme and increased emissions of CO (factor of 2 for extra-tropical fires and 1.5 for savannahs) following Petrenko et al. (2017), which has successfully reproduced the satellite observations of AOD with a series of adjustments to biomass burning emissions."

*L341-342: The match between 900 and 800 hPa looks like a coincidence to me, as the model profile is highest there but it is a local minimum in the observed profile. Can you argue that I should have more confidence in the match in that region?*

That sentence has now been removed because we discuss this comparison in more detail in the preceding paragraph in response to an earlier comment.

*L343-350: This "limitations and future work" paragraph would probably fit better in Section 4.*

We have moved this text to Section 4 as suggested.  It does fit better there. Thank you for that suggestion.

*L358-359: For all of these "in prep" references, it would be good to either add more detail to this paper or provide a better reference, such as to a conference presentation if one exists.*

The parameterization paper (Val Martin and Kahn, 2018) has been submitted to *Remote Sensing*. A copy has been uploaded with our responses. We have also added substantially more detail in Section 2.2 so that readers mainly interested in the GEOS-Chem implementation do not necessarily need to have the Val Martin and Kahn paper nearby to understand this paper.

*L409: But there is a large negative change in Northwest Canadian Figure 6a, which does not seem consistent with you OH-based explanation. What is the cause of that feature?*

Features like that are not inconsistent with our explanation. Very close to fires, the injection height will directly impact the CO. The CO lifetime against oxidation by OH is on the order of a month. In Figure 6a, the blue grid boxes indicate that at those particular locations, there is less CO in the "new injection scheme" model than the standard model in the 510 hPa model layer. The way emissions are injected varies by grid cell. In some locations our scheme does not move emissions up in the atmosphere. In the following figure included here in the response, we show the emitted fractions from both the standard model (blue) and new injection scheme (red) at the darkest blue grid cell in Northwest Canada of Figure 6a. The new injection scheme only emits 2% at around 700 hPa, while the standard model emits a large fraction close to the boundary layer top. Thus, it is totally possible that the model convection brings more CO upward in the standard model and the CO from new injection scheme is less than that from the standard model at 510 hPa. This also can be seen in Figure 3. The percentage of BB emissions injected above 700 hPa is really variable in that area.

[Figure]

*L425-430: Rather than referring to "most" stations with no change and "some" with decreases, can you be quantitative? How many sites had no change (and does this mean a change of less than 1 ppb? 10 ppb?), and how many had decreases? Did any have increases?*

Thanks for the comments. We modified the texts to be more quantitative, and in response to an earlier comment, this information is now in Section 2.5.3.

**"2.5.3 Surface Observations**
Leung et al. (2007) showed that the choice of injection height for boreal fire emissions impacts the simulation of surface CO mixing ratios in the Northern Hemisphere. They compared GEOS-Chem simulated anomalies in CO mixing ratios with surface measurements from the NOAA ESRL Global Monitoring Division (GMD), Carbon Cycle Cooperative Air Sampling Network (Novelli et al., 2003). Therefore, we also performed a comparison with monthly mean observations from 18 sites that may have been impacted by fires during 2008. In most locations (16 of 21) where we conducted comparisons, the model with the MISR-based injection height did not produce notably different surface monthly mean CO mixing ratios (i.e. changes are less than 1 ppb). However, there are four stations where the updated model produces substantially lower monthly mean surface CO mixing ratios than the standard model, and this change produces a better simulation of CO at these locations. We present these results in Section 3.3."

*L467-469: You don't present any comparisons to satellite observations in this paper, so this doesn't really belong in your summary.*

Actually, we did comparisons to satellite observations. We didn't show them because of the problem noted here. Observations with low vertical resolution are not helpful in this context. We think it is important to mention this in the paper so that others can avoid extra work if possible.

*Typos and Style Suggestions*

*L98: Replace "this trace species" with "PAN" to be clear.*

Thanks. Corrected.

*L134: I'd suggest that "account for" is closer to what you mean than "acknowledge" here.*

Thanks. Corrected.

*L137: Extra "s" after "biome"*

Thanks. Corrected.

*L330: Extra comma after "specific"*

Thanks. Corrected.

*L352: I don't think you need the word "above" as you mention the section number.*

Thanks. Corrected.

*L371-372: You discuss the results in Figure 8c before introducing the figure. I'd cut the previous sentence to ". . .emissions pushed higher in the atmosphere."*

Thanks. Corrected.

*L394: I'm not sure "understand" is the right word here. Maybe "simulate"?*

Thanks. Corrected.

*L398-399: Instead of referring to the "example in Figure 6", I'd suggest describing it as "the 4 July smoke plume from ARCTAS (Fig. 6)".*

Thanks. Corrected.

*L412: Word "typically" is redundant with "Typical" at the beginning of the sentence.*
Thanks. Corrected.

*L424: Extra space after "Network"*

Thanks. Corrected.

*L463: I'd suggest changing this to "provided access to CO profiles that could be used for model-measurement comparison"*

Thanks. Corrected.

*L464: "do not appear", instead of "to not appear"*

Thanks. Corrected.

**Short comments #1**

*Dear Authors,*

*firstly, I asked the editorial team to change the paper type of your manuscript, as you are clearly describing a new development this should be a "Development and technical" paper and not an "Evaluation paper". A "Developement and technical paper" of course has to include an basic evaluation of the new development, but an evaluation paper does not really contain larger new developments. As a consequence, please provide the version number of GEOSChem model and a name (Acronym and version number of the newly developed scheme) in the title of the article. The title could look like:*

*"Development and implementation of a new biomass burning emissions injection height scheme (BBEIH vx.y) for the GEOSChem model (version z.g)"*

Thanks for the suggestion. We changed our title to: "Development and implementation of a new biomass burning emissions injection height scheme (BBEIH v1.0) for the GEOS-Chem model (v9-01-01)".

*Secondly, the paragraph you named "Data Availability" should be named "Code and data availability". As explained in https://www.geoscientific-model-development.net/about/manuscript_types.html. GMD is encouraging authors to upload the program code of models (including relevant data sets) as supplement or make the code and data of the exact model version described in the paper accessible through a DOI (digital object identifier). In case your institution does not provide the possibility to make electronic data accessible through a DOI you may consider other providers (eg. zenodo.org of CERN) to create a DOI. Please note that in the code availability section you can still point the reader to how to obtain the newest version. If for some reason the code and/or data cannot be made available in this form (e.g. only via e-mail contact) the "Code Availability" section need to clearly state the reasons for why access is restricted (e.g. licensing reasons).*

*Especially, please note, that it is not enough, that the code will be available in the future. It must be available now and the exact version of the code published in this article needs to be made available.*

*Additionally, please note, that the exact code needs to be available to the Editor and the Referees for the review process.*

*Yours, Astrid Kerkweg*

Thanks for the comments. Our code and data were submitted as supplemental materials.